# Sample-Efficient Mapspace Optimization for DNN Accelerators with Bayesian Learning

Grace Dinh*, Iniyaal Kannan*, Hengrui Luo†, Charles Hong*
Younghyun Cho*, James Demmel*, Xiaoye Sherry Li†, Yang Liu†
*UC Berkeley. *dinh,iniyaalkannan,charleshong,younghyun,demmel@berkeley.edu*
†Lawrence Berkeley National Lab. *hrluo,xsli,liuyangzhuan@lbl.gov*

*Abstract*—**Achieving high performance for machine learning domain-specific accelerators requires the careful choice of a mapping from the algorithm to the accelerator. Most algorithms for finding mappings either optimize over a coarse performance *model* or experimentally evaluate the performance of a large number of different mappings on the space. However the number of samples required by these empirical models can be prohibitive in settings (e.g. when using cycle-accurate simulators) where evaluations are expensive.**

**This paper evaluates the use of Bayesian optimization based approaches for finding mappings for hardware accelerators in settings where high sample efficiency is required. Our approaches converge to mappings comparable to that of Timeloop's mapper while requiring an order of magnitude fewer iterations. Furthermore, our method produces surrogate models that can be used for transfer learning to new hardware configurations, further reducing the sample complexity by roughly a factor of $2$.**

## I. Introduction

Domain-specific hardware accelerators have become increasingly important in enabling efficient, performant execution of linear algebra and machine learning applications. However, attaining good performance on accelerators requires a careful choice of a *mapping* describing how the algorithm is to be executed on the target accelerator. These mappings, which encompass choices such as tiling dimensions, loop ordering, and *spatio-temporal mappings* (i.e. deciding which axes to map to accelerator parallelism), can significantly affect performance by up to four orders of magnitude [17].

However, the space of possible mappings (*mapspace*) is challenging to search, as the number of choices that comprise a mapping leads to a combinatorial explosion in the number of possible mappings. Furthermore, this space is highly nonconvex and changes significantly with the target architecture; as a result, it is desirable for mapping methods to be efficiently generalizable across a wide variety of hardware parameters and architectures, especially in settings such as hardware-software codesign where efficient mappings must be computed for a wide variety of both algorithmic and hardware targets.

In order to handle complexity of searching over a high-dimensional mapspace, many approaches rely on *performance models* that are either mathematically simple enough to be optimized over [10], [37] or cheap enough to be queried thousands or tens of thousands of times [25] in brute-force approaches. However, as we show in Section II, these models often diverge from actual performance significantly, limiting their effectiveness.

Alternatively, *feedback-driven* approaches, which iteratively search over the mapspace using black-box search (e.g. genetic algorithms or reinforcement learning) or gradient descent, can be guided not only by performance models but also by measured or simulated performance. However, many previous methods are extremely sample-inefficient, sometimes requiring millions of samples to train models [8], and have difficulty generalizing to new hardware targets or problem dimensions [13]. This poses unique challenges for the case of hardware design-space exploration (DSE), where mappings must be computed for a variety of hardware targets.

This paper explores the use of Bayesian optimization to find performant mappings in a *sample-efficient* manner, while constructing *surrogate functions* for performance that can be efficiently queried and optimized. We then use these surrogate functions to perform *transfer learning* across different hardware configurations, showing that, in contrast to previous gradient-based approaches [8], [13] and reinforcement learning approaches [12], our approach can generalize to hardware configurations not in its training set. Our approach also provides automatic problem-specific sensitivity analysis for mapspace parameters during optimization.

## II. Background

Algorithms for finding performant mappings generally fall into one out of three categories:

- *Heuristics* perform one-shot analytic optimizations over a performance model (either defined explicitly to be used as an optimization target, or embedded implicitly in the heuristic). Such methods include polyhedral models [1], [7], [18] and constrained-optimization based approaches [10], [36]. While heuristics are efficient and generalize easily across hardware parameters and problem sizes, they are often limited to optimizing a subset of the mapspace (e.g. tilings and reorderings only, as in [24]), and rely on an analytic model which can only coarsely approximate performance.
- *Random search* methods [5], [25], [35] use brute force to sample and evaluate a large number of points in the mapspace. However, the size of the mapspace (well over

$10^{20}$ points [26]) necessitates a large number of samples to achieve good performance.

- *Feedback-driven methods* use statistical or ML methods to iteratively explore the mapspace. These approaches include black-box optimization techniques such as genetic algorithms [12], [14], reinforcement learning [34], and Bayesian optimization [29], which aim to require fewer samples than brute-force methods. Alternatively *gradient-based methods* [8] build differentiable surrogate functions that estimate performance based on input parameters. However, training gradient-based methods require millions of samples to build surrogate models, and the resulting surrogates cannot easily be used for hardware architectures not yet seen even after fine-tuning, and must be fully retrained from scratch at nontrivial expense [13].

The aforementioned methods tend to be heavily reliant on performance models to attain good results. This reliance is explicit in the case of heuristic-based approaches that optimize objectives that model performance. On the other hand, the reliance on efficient models is implicit for random search and many feedback-driven methods: their reliance on thousands or even millions of performance samples renders them impractical for use when the cost of performance evaluation is high, e.g. when using cycle-accurate RTL emulators such as FireSim [15], which can take several minutes to run a single neural net layer on a standard AWS FPGA. This is further exacerbated in the hardware DSE setting, where performance evaluation must be performed not only for each mapping but also for each potential hardware backend. As a result, these methods are often run using fast analytic models such as Timeloop [25] and Maestro [19] instead.

However, such analytic models can diverge from actual performance significantly [33]. To see the significance of this difference, we generated 2000 randomly chosen mappings for a variety of convolutions and matrix multiplication problems. We then evaluated the cost of these mappings, first by executing the code corresponding to these mappings on the GEMMINI [6] DNN accelerator running on the cycle-accurate Firesim [15] RTL emulation platform, then by using an inexpensive analytic performance model, Timeloop [25]. Hardware parameters (memory bandwidth/sizes, systolic array dimensions, etc.) were identical for both targets. Figure 1 shows a log-log scatter plot of the ratios of the cycle counts generated by Timeloop and Firesim, which can differ by up to two orders of magnitude.

As a result, we wish to develop feedback-driven methods for finding performant methods with *high sample efficiency*. Furthermore, we would like our approach to *generalize* inexpensively to new hardware configurations. Our approach is to to use Bayesian optimization, which has been shown to be effective for optimizing complex functions with a limited number of evaluations due to its faster convergence and ability to handle multiple parameters. One of its key strengths is the construction of *surrogate models* using Gaussian processes, which are powerful tools for modeling complex interactions and potentially noisy functions. Bayesian approaches have

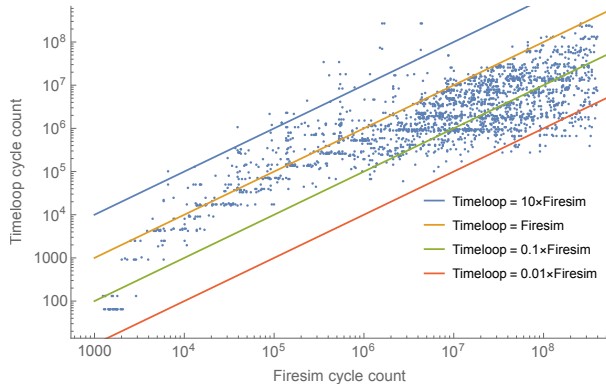

Fig. 1. Convergence of GPTune and Timeloop's random-search mapper

been used to perform black-box optimization in domains where sample efficiency is paramount, including algorithm optimizations on supercomputers [4], [21] and optimizing hardware parameters for accelerators [23], [27], [34]. We apply similar techniques to optimize over *software* mapspaces for accelerators, improving on previous attempts to do so [30] by using an efficient encoding scheme to embed mapping parameters into a mathematical space that can be more easily searched.

This paper considers mapspaces consisting of tile sizes and loop orderings (dataflows) for multilevel memory hierarchies. Our techniques directly generalize to mapspaces including spatio-temporal mappings as well; we leave benchmarking those to future work.

### A. Mapspace Encoders

Bayesian optimization assumes an objective function (in this case, a performance metric such as latency, cycle count, or energy) that takes as input a set of numerical, usually continuous, variables. However, decisions that comprise a point in the mapspace, such as loop orderings, are discrete. These discrete variables fall into one out of two categories.

*Discrete numerical variables*, such as tile sizes, are mostly integral. However, depending on the hardware target, their discreteness may take the form of a requirement to be a multiple (e.g. of the size of a vector unit) or factor (e.g. of the problem size, to allow for perfectly nested loops without tail cases) of some integer. We represent such variables as continuous variables in the optimization program and round them in order to find the actual mapping parameters; such rounding approaches have experimentally been shown to match or exceed discrete surrogate based approaches [16].

*Categorical variables*, such as loop orderings, are members of a finite, unordered set. These variables can be dealt with in one of two ways:

- by *directly applying surrogate-based optimization approaches* to them. Handling categorical variables in optimization, especially in Bayesian optimization, poses unique challenges due to their discrete and unordered nature, especially in domains comprised of both continuous and categorical variables. Several approaches for

integrating the continuous and categorical optimization methods have been studied, including one-hot encoding [31], bandit models [28], and hybrid Monte Carlo tree search [21]. However, the combinatorial complexity of the mapping problem complicates such approaches; for instance, a batched convolution with seven nested loops has $7! = 5040$ possible loop orderings per memory level, and general categorical optimization methods are unable to take problem-specific information that could guide search over this space into account (for instance, that performance is likely to be changed less by swapping the order of two loops than reversing the entire loop nest).

- by creating a *mapspace-specific encoding* from continuous variables. For example, for loop orderings, we optimize over *scores* for each axis and order the axes from lowest to highest score. A similar approach, as in [20], can be used for spatio-temporal mappings.

**Dealing with hardware constraints.** The set of valid mappings is bound by a set of constraints, which we will address in this section by developing a mapping from an *unconstrained* feature space $\mathbf{f} = (0, 1]^d$ (for some integer $d$), which can be easily optimized over, to the set of valid mappings. We will denote axes of this feature spaces as $f_i$.

Many constraints are simple constant bounds on the numeric variables (for instance, ensuring that tile sizes must be smaller than the sizes of the data tensors) and can be dealt with by scaling the variables appropriately. For instance, instead of optimizing a loop tile size $t$ under the constraint that it is bounded above by the size $s$ of the input problem, we can instead optimize a value $f \in (0, 1]$ and set $t = sf$.

However, other constraints may result in more complicated inequalities. For example, consider a 2D convolution with $b$ batches, $c$ input channels, $k$ output channels, and windows of size $r \times s$, outputs of size $w \times h$. If we wish to tile these axes in such a way that the tiled inputs and weights can fit into a scratchpad of size $M$, the tile sizes $t_b, t_c, ...$ must satisfy the following constraint:

$$t_c t_k t_r t_s + t_b t_c (t_w + t_r)(t_h + t_s) \leq M \qquad (1)$$

Rejection sampling is often used to handle such constraints, but has two drawbacks. First, setting an objective value to assign to invalid mappings is a nontrivial hyperparameter optimization problem; an overly high value can cause unwanted behavior in a learned surrogate function (leading to unpredictable behavior when, for instance, doing transfer learning), while an overly low value may not be enough to discourage the optimizer from considering invalid maps. Furthermore, the rejection probability can be high - increasingly so as the dimensionality increases - significantly driving up the number of iterations required. In fact, prior work [30] requires sampling $22K$ points in order to produce $150$ valid mappings, which drastically increases the cost of this approach.

As a result, our goal is to develop a mapping from every point of $\mathbf{f}$ to a *valid* point in the mapspace. Since all nontrivial constraints in the mapspace take the form of capacity constraints similar to that of (1) over the tile sizes [10], we instead

optimize the *aspect ratio* of the tiles, and then scale all the tile sizes by the same factor to maximize memory utilization. We believe this approach also improves the ability of the learned model to generalize across problem sizes, as communication-optimal tiles for many problems such as matrix multiplication [11] retain the same aspect ratio (square tiles) as long as problem sizes are sufficiently large.

More concretely, consider the memory constraint given in (1). Instead of directly optimizing over the tile sizes $t_{b,c,...}$, we optimize the variables $f_{b,c,...} \in (0, 1]$, which we scale by a *common* multiplier $\alpha$ to obtain

$$t_{b,c,...} \approx \alpha f_{b,c,...} \qquad (2)$$

In order to determine the value of $\alpha$, notice that substituting (2) into (1) gives the following inequality:

$$\alpha^4 \left[ f_c f_k f_r f_s + f_b f_c (f_w + f_r)(f_h + f_s) \right] \leq M \qquad (3)$$

As there is no reason not to maximize memory utilization, we replace the inequality with equality, which therefore provides the value of $\alpha$:

$$\alpha = \left( \frac{M}{f_c f_k f_r f_s + f_b f_c (f_w + f_r)(f_h + f_s)} \right)^{1/4}$$

We can then round the resulting values of $\alpha f_{b,c,...}$ down to the nearest valid value (to satisfy discreteness and maximum tile size constraints) of $t_{b,c,...}$, ensuring that each point $\vec{f} \in (0, 1]^d$ corresponds to a valid mapping.

## III. EVALUATION

For our experiments, we optimize for energy cost on a hardware model based on GEMMINI [6] with a four-level memory hierarchy: a register, an accumulator for the outputs, a scratchpad for the inputs and weights, and DRAM. We test our mappings on Timeloop [25], which takes as input an algorithm and a hardware configuration and provides (1) an analytic performance *model* for energy and latency and (2) a pruned random search based *mapper*. While our approach is designed to target hardware models with far higher per-sample cost than than Timeloop's model, we use Timeloop in order to allow for the use of its random-search mappers (which would be infeasibly expensive if run with a cycle-accurate simulator) as a comparison target. We leave benchmarking on an (expensive) cycle-accurate simulator and comparing performance to model-based (brute-force and heuristic) mappers to future work.

To perform Bayesian optimization, we use GPTune [3], an autotuning suite designed for optimizing applications by utilizing Bayesian approaches. GPTune incorporates multi-task learning and transfer learning algorithms to share knowledge of obtained performance samples among multiple tasks, improving tuning results. It enables quick prediction of optimal tuning parameters for new tasks using data from existing tasks. Additionally, GPTune supports multi-objective tuning, hybrid models [21] for mixed categorical and continuous variables, and non-smooth objective tuning [22].

In order to reduce statistical variance, all experiments were averaged over three independent runs.

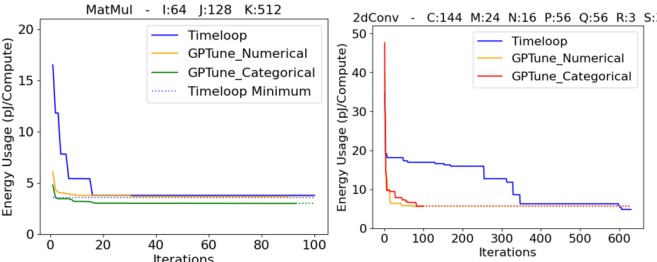

Fig. 2. Energy (lower is better) attained by Timeloop's brute-force mapper and GPTune for matrix multiplication (left) and 2D convolution (right). GPTune was run for 100 iterations; the best value found is indicated with dotted line extending to the right.

## A. Convergence

Figure 2 shows the energy consumption of the best mapping found so far at each iteration, comparing Timeloop and GPTune (we run 100 iterations for GPTune).

For GPTune, we show results for both approaches to optimizing over categorical variables (in this case, loop orderings) described in Subsection II-A. We note that directly embedding loop orderings into the problem produces superior results to the score-based approach for matrix multiplication but inferior results for convolutions, likely because of the higher dimensionality of convolutions compared to matmuls: a 3-nested loop matmul leads to roughly $(3!)^4 = 1296$ choices for loop orderings over the four levels of the memory hierarchy, while the 7-nested loop convolution results in roughly $(7!)^4 \approx 6e14$ choices. This suggests using categorical encodings works well for relatively low-dimensional problems, whereas score-based encodings are better for higher-dimensional problems.

For matrix multiplication, GPTune converges in roughly 20 runs to 2.98 pJ/compute, a value $16\%$ better than the 3.56 pJ/compute that Timeloop achieves after $4000$ runs (note that Timeloop plateaus after an average of $420$ iterations).

For 2D convolutions, GPTune converges in (on average) 50 iterations to a minimum of 5.26 pJ/compute, a value that it took an average of 627 iterations for Timeloop to beat. Furthermore, after 4000 iterations, Timeloop's best value was 4.32 pJ/J, roughly $17\%$ better than GPTune's.

## B. Transfer Learning

In many settings, such as hardware DSE, the ability to leverage data collected on one or more hardware configurations to guide search on a hitherto unseen hardware configuration can prove useful. However, support for transfer learning across hardware configurations has proven limited so far. Random search and many black-box optimization algorithms, such as genetic algorithms (e.g. GAMMA [12]) do not support transfer learning and must be run from scratch for every hardware target and algorithm. Attempts to apply the differentiable surrogate models found by Mind Mappings [8] to hardware architectures not in the training set resulted in performance one to two orders of magnitude worse than running Timeloop's random mapper and GAMMA from scratch.

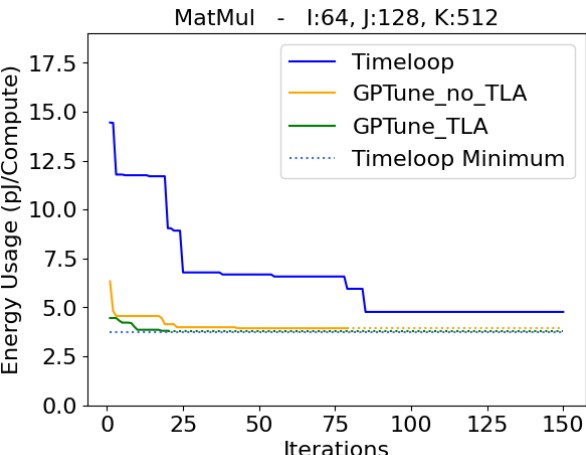

Fig. 3. Transfer learning to a new (not in training set) hardware configuration for matrix multiplication, compared to GPTune with no prior knowledge and Timeloop.

Previous Bayesian optimization based approaches to mapspace search [30] have not considered transfer learning.

The Gaussian process surrogate models produced by GPTune possess the capability to facilitate transfer learning. We first train a surrogate model taking into consideration both task parameters (i.e., tensor dimensions) and hardware parameters (i.e., memory hierarchy specifications), utilizing GPTune's multitask learning algorithm for four distinct memory hierarchy configurations. Subsequently, we refine this model for 20 iterations, employing the target hardware configuration that was absent from the initial training set.

Figure 3 shows the transfer learning, which converges to a mapping providing 3.81pJ/compute (on par with an uninitialized GPTune) in 10 iterations (roughly half that of an uninitialized GPTune). This figure requires Timeloop an average of 1600 iterations to beat.

## C. Sensitivity Analysis

The surrogate models can be used for sensitivity analysis as well, by applying Sobol analysis [32] to attribute the part of the variance of the output can be attributed to each of the inputs. We leverage GPTune's sensitivity analysis interface, which internally invokes SALib [9] for computing Sobol indices from the trained surrogate model. For matrix multiplication, the most important axes were the tilings of the $64 \times 512 \times 128$ matrix multiplication example shown in Figure 2, the most important axes were the tilings of $k$ at the register and accumulator levels, and the tiling of $j$ at the register level; the surrogate model was several orders of magnitude more sensitive to the tiling parameters than the loop ordering ones, which lines up with previous work [13], [35] showing that tilings are the most important mapping parameter.

For high-dimensional problems such as convolutions, we believe this surrogate model may be used to perform automated dimension reduction - perhaps even during the optimization process itself; we leave this to future work.

## IV. CONCLUSION, DISCUSSION, AND FUTURE WORK

This paper demonstrates the feasibility of using Bayesian optimization to perform mapspace search using very few (under 100) samples, which can be reduced even further by using transfer learning from data collected for other hardware configurations. Key to our approach is the construction of an encoding scheme that ensures that every point in the search space given to the Bayesian algorithm corresponds to a valid mapping. The clearest application of this approach is to settings where the cost of measuring performance data is expensive, such as running code on cycle-accurate simulators, especially in the context of hardware design-space exploration. The construction of compiler infrastructure required to test the performance of our approaches on these simulators is currently ongoing.

It may also be interesting to extend the Bayesian multitask learning approaches for transfer learning across hardware *configurations*, as seen in this paper, to transfer learning across different hardware *simulation fidelities*. Using a large number of samples from a cheap but coarse model to guide search over a much more expensive space may allow for exploration of a large portion of the mapspace without wholly relying on the accuracy of performance models.

In a similar vein, we wish to experiment with the use of analytic one-shot models, such as CoSA [10] and theoretically optimal tiling methods based on Brascamp-Lieb inequalities [2], [24]. While these methods are reliant on analytic models for performance and may not be optimal in practice on real hardware, they can provide cheap initial data points that may help to significantly accelerate search.

We are also investigating more sophisticated approaches for high-dimensional Bayesian optimization that combine multiple techniques that are tailored to various application domains.

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
