# OpenReview forum: "Sample-Efficient Mapspace Optimization for DNN Accelerators with Bayesian Learning"
_iscaconf.org/ISCA/2023/Workshop/ASSYST — ASSYST Oral_

### Official Review · Reviewer_A6bB · 2023-05-05
**"Title: Sample-Efficient Mapspace Optimization for DNN Accelerators with Bayesian Learning" This paper is about "mapping" - i.e., decomposing an algorithm to hardware instructions- targeting machine learning applications. The feedback-driven methods are explored, and the Bayesian optimization is considered using the Gaussian processes for surrogate functions.**

**Rating:** 5
**Confidence:** 3

**Review:**

A clear definition of mapping is not in the Abstract. I am not sure about making two paragraphs of the Abstract. In general, the Abstract was prepared with some vague concepts, such as Timeloop's mapper, mapping, new hardware configurations, sample complexity, etc.

In the Introduction Section, the definition of mapping occurs. The classification of mapping approaches is good and structural.

For Fig.1, I highly recommend explaining the concepts clearly, such as cycle count. The reader cannot understand details on the mapping; what do these FireSim, etc., simulators do? What are the details of hardware mapping? Up to page 2, the reader only knows this:
"However, attaining good performance on accelerators requires a careful choice of a mapping describing how the algorithm is to be decomposed into hardware instructions that can be run on the target accelerator."

In general, I liked the references in the paper. The numbers are sufficient and properly placed for each piece of information. Authors might consider preparing a benchmark/survey for their future work. (This comment was written before reading the author(s)' sentence: "...we leave benchmarking for those to future work." So, it is good to see this information.)

In Section II., like the overall issue in the manuscript, the majority of abstract concepts make reading harder. For instance, before sub-section A., the final sentence, "Our techniques directly generalize to mapspaces including spatio-temporal mappings as well; we leave benchmarking for those to future work." Here, "the spatio-temporal mappings" is an abstract concept.

While reading the paper, some questions are better to answer before (maybe in the Introduction), with a more structured manner (like a Table, nomenclature, etc.): What are the constraints and optimization parameters for the "mapping"? What are the inputs (general) that the optimizer/simulator takes? In which format? And what are the outputs? What kind of automatic and user-defined parameters that these mappers take into account?

The end of Section II. is the most crucial part of the manuscript's contribution. This part was expected to be more explicit. For instance, what does Equation 1 stand for? Maybe a figure explains it well. The main contribution is blurry and the reader looks for the "This is the main contribution..." sentence to explain the novelty of the manuscript. What does the alfa stand for?

Please see the following language issue: "While our approach is designed to target hardware models with far higher per-sample cost than than Timeloop’s model," --> There are two "than"s.

The experimental section is limited and it only addresses a selected architecture with some size: 64×512×128 (why is it so important as it was indicated in the paper?)

There is a discussion and future work but I expect to see the conclusions.

**Review (Strengths/Weaknesses):**

WEAKNESSES:
-The article assumes that readers have knowledge of abstract concepts, and therefore fails to explain some essential concepts. In my general comment above, some examples were provided to authors correspondingly; some concepts require clarification.
-Another weakness is that the contribution made by the research is not explicitly stated, leaving a question mark about the exact scientific contribution.
-The algorithmic flow used for optimization is not fully described. There is no mathematical discussion of how certain scientific concepts such as Bayesian, Gaussian, and Surrogate Model are applied.
-The introduction refers to a "mapping describing how the algorithm is to be decomposed into hardware instructions," but there is no explanation of the "instructions".
-The article has a discussion and future work section, but there is no Conclusion Section.

STRENGTHS:
-The selected problem is an interesting one.
-The references chosen are appropriate.
-The hierarchical mapping organization and reference arrangement in the early sections are suitable. As such, the article could be expanded into a benchmark paper.

Overall, this study would be better suited as a benchmark paper focusing on mapping tools. However, in its current form, without revisions to address the weaknesses mentioned above, it does not appear to provide a thorough scientific contribution. (It is possible to consider it as a benchmark paper if authors revise the weaknesses and general comments above.)

**Reviewer Expertise:**

Knowledgeable: I used to work in this area and/or I try to keep up with the literature but might not know the latest developments.

---

### Official Review · Reviewer_BUFA · 2023-05-06
**Sample-Efficient Mapspace Optimization for DNN Accelerators with Bayesian Learning**

**Rating:** 6
**Confidence:** 3

**Review:**

The paper proposes a method for optimizing the mapping of machine learning algorithms on specialized hardware accelerators. It utilizes Bayesian optimization and Gaussian processes to explore the space of possible mappings and constructs surrogate models to estimate energy cost. Results show that the proposed method consistently finds performant mappings in far fewer iterations than random search. It also produces surrogate models that can be used for transfer learning across different hardware configurations. The paper also discusses how these models can be leveraged to identify important parameters for optimization.

**Review (Strengths/Weaknesses):**

Strengths
+ With the architecture of accelerators and ML workloads changing rapidly, finding an efficient mapping quickly that performs well is a challenging problem. The use of Bayesian optimization is an effective method for finding an efficient mapping with a small number of evaluations, especially since the search space is huge.
+ The method intelligently handles discrete numerical variables (tiling size) and categorical variables (loop order) by representing them as continuous variables or by creating a mapspace-specific encoding respectively.
+ The combinatorial complexity of the mapping problem is complicated by categorical variables with large and unordered sets, but this complexity reduces since categorical values are only used for loop order, and for ML workloads the number of loops are fixed.
+ The method is versatile, and the resulting surrogate models can be extended via transfer learning to other unseen hardware configurations.


Weaknesses
- While the results suggest that the proposed Bayesian optimization approach performs well against a random search method, the paper lacks a comparison of the proposed approach with other state-of-the-art Feedback-driven methods for optimizing hardware-software mapspaces.
- The success of mapspace encoders is highly dependent on both the quality of the encoding scheme and the optimization algorithm employed. More details (apart from citations) such as the methodology behind the score determination used in the study would be informative.
- The related work could improve by discussing similar prior work and how the proposed work differ from these prior works such as “Using Bayesian Optimization for Hardware/Software Co-Design of Neural Accelerators”
- The paper has some typos
    ------> Black-box methods, such as reinforcement learning *[cite]*.....
    ------> ensuring that tile sizes must be smaller sizes of the data tensors) and can be dealt with by scaling *the* the variables appropriately.
    ------> We can *then* then round the resulting values of αab,c,... down to the nearest


**Reviewer Expertise:**

Knowledgeable: I used to work in this area and/or I try to keep up with the literature but might not know the latest developments.

---

### Official Review · Reviewer_YwLu · 2023-05-10
**The paper is not fit for presentation at the workshop in its current form.**

**Rating:** 4
**Confidence:** 3

**Review:**

There are several problems with the paper that hinder understanding of the proposed research.

- The paper does not pinpoint the problem clearly. Many performance modeling approaches exist in prior literature. What is the key benefit of the proposed approach?

- The evaluation is very weak. The details of the hardware model (beyond assumptions about memory hierarchy) are missing. Provide more details in Section III.

- The evaluation needs stronger case studies.

- The meat of the paper (Section II) is extremely thin. Expand on the details.

- Many typos. Figure captions and axes not readable.

**Review (Strengths/Weaknesses):**

The main problem is that the contribution is very thin and in relation to prior work it is not clear what is the novelty.

The problem is however timely, but the contribution here is very minimal.

**Reviewer Expertise:**

Little or no familiarity.

---

### Official Review · Reviewer_eLwc · 2023-05-10
**Investigating Bayesian Optimization for Mapping DNN Accelerators: A Promising but Limited Study**

**Rating:** 5
**Confidence:** 3

**Review:**

This paper addresses the challenge of mapping machine learning algorithms onto domain-specific architectures due to the complexity arising from the numerous possible mappings and the diverse target architecture. It focuses on feedback-driven mapping groups and examines the use of Bayesian optimization to explore the mapspace. The paper acknowledges that previous feedback-driven approaches may diverge from actual performance.

- The paper offers a well-structured overview of the target problem and previous solutions, but lacks a clear problem statement specific to this paper, hindering the understanding of the proposed optimization's rationale.
- Figure 1 contains vital information for understanding the problem statement, but several details and underlying experiment settings are missing, making it difficult to follow.
- Section II contains few paragraphs directly explaining the paper's methodology, which are obscured by more general information. From this section, readers only gain two high-level insights: (i) the focus on mapspaces with tile sizes, loop orderings, and multilevel memory hierarchies; and (ii) the goal to develop a mapping between the tuning parameter space and mapspace.
- Due to the complexity of mapping diverse algorithms and architectures, the limited experiments do not adequately demonstrate the effectiveness of Bayesian optimization.
- Some other issues of the paper are poor writing, typos, disorganized paragraphs, and small figure fonts.

**Review (Strengths/Weaknesses):**

- The addressed problem is highly relevant, given the rapidly evolving ML algorithms and hardware accelerators.
- The paper presents motivating experiments (e.g., Figure 1) that help readers understand the limitations of current approaches.
- Despite limited evaluation, the study provides interesting observations and insights based on metrics like energy consumption, although clearer takeaways would be beneficial.

**Reviewer Expertise:**

Knowledgeable: I used to work in this area and/or I try to keep up with the literature but might not know the latest developments.